# Brief communication: An elliptical parameterisation of the wind direction rose

Edward Hart[1]

[1]Wind Energy and Control Centre, Department of Electronic and Electrical Engineering, University of Strathclyde, Glasgow, United Kingdom

**Correspondence:** Edward Hart (edward.hart@strath.ac.uk)

**Abstract.** This brief communication presents a parametric model for the wind direction rose, based on ellipse geometry. Such a model supports standardisation and identification of generally representative cases, while also enabling systematic analyses of wind rose "shape" impacts on the benefits of proposed wind farm design and control innovations. Formulations include analytical wind direction rose modelling, model fitting to measured data via gradient descent minimisation of sum-of-square-errors, and goodness-of-fit measures. Testing on wind direction data from real offshore wind farms confirms good performance, indicating this parametric model is useful to wind energy research and development efforts. Possible model extensions are also discussed, including their benefits and drawbacks.

## 1 Introduction

At the wind farm scale, the direction of the wind strongly affects the levels of turbine-turbine interactions and, hence, the flow conditions experienced by individual turbines (Meyers et al., 2022; Dallas et al., 2023). This, in turn, impacts energy production, optimal turbine layout, the efficacy of different wind farm control strategies, and turbine reliability (Slot et al., 2018; Amiri et al., 2019; King et al., 2021; Stanley et al., 2023). The wind direction rose is therefore of broad importance to wind farm design and operation. Despite this, to the best of the authors knowledge, a standard parametric model for the wind direction rose is yet to be adopted by the industry. The benefits of such a model would include: 1) allowing for standardisation and supporting the identification of typical or generally representative cases[1] 2) enabling the influence of wind rose "shape" on energy yield and reliability impacts resulting from proposed innovations to be systematically explored. For example, recent studies on wake steering and turbine layout optimisation (King et al., 2021; Stanley et al., 2023) report increased energy yields obtained for single candidate wind roses at each modeled wind farm. The robustness and generality of these studies would be enhanced if, instead, yield increases were determined across a range of wind roses, obtained by systematically varying model parameters. While various circular distributions have been proposed in the literature, their complexity, difficulty of fitting, and relatively large parameter-sets render them impractical for such applications (Mardia and Jupp, 2009; Kim and SenGupta, 2021; Yang et al., 2022; Yang and Dong, 2024). This brief communication therefore proposes a novel and simple parametric model for the wind direction rose, utilising ellipse geometry to minimise the required number of model parameters.

---

[1]For example, we see this for wind speed distributions, where a Weibull distribution with shape parameter close to 2 is common (Shu and Jesson, 2021).

Section 2 develops the parametric wind direction rose model, along with necessary theory to allow for fitting to measured data and evaluation of goodness-of-fit. Example implementations are then provided in Section 3. Use cases and possible model extensions are discussed in Section 4, before Section 5 concludes the paper.

## 2   Methodology

### 2.1   Standard ellipse equations

The ellipse, which generalises the concept of a circle, has the following standard parametric form,

$$(x, y) = (a\cos(\theta), b\sin(\theta)) \quad \text{for } 0 \leq \theta \leq 2\pi,$$

where the ellipse's axes (of length $2a$ and $2b$) align with the coordinate system's $x$ and $y$ axes, respectively. More generally, if the ellipse is rotated by an angle $\phi$ while remaining centered at the origin, its parametric form becomes,

$$\begin{aligned}
(x, y) \quad = \quad & \big( a\cos(\theta)\cos(\phi) - b\sin(\theta)\sin(\phi), \\
& a\cos(\theta)\sin(\phi) + b\sin(\theta)\cos(\phi) \big) \\
& \text{for } 0 \leq \theta \leq 2\pi,
\end{aligned}$$

Note, any ellipse of the latter form may be reduced to the former via a simple change of coordinates. Without loss of generality we therefore focus on the first parametric form. The total area enclosed by an ellipse is,

$$A = \pi a b.$$

### 2.2   Ellipses of unit area

Imposing the restriction $A = 1$ it follows that,

$$b = \frac{1}{\pi a},$$

and, hence, the number of parameters defining the ellipse is reduced to one, i.e. just $a$. An ellipse of unit area may be naturally interpreted as a wind direction rose by defining the probability of the wind blowing from between directions $\theta_1$ and $\theta_2$ to be equal to the area enclosed by the ellipse between those two angles. As such, a formula for ellipse segment areas is required. This may be obtained by observing that an ellipse with semi-major axis $a$ (aligned with $x$) and semi-minor axis $b$ (aligned with $y$) becomes a circle if the $y$-axis is scaled by a factor of $a/b$ (see Figure 1). This scaling effects the segment angles $\theta_1$ and $\theta_2$, resulting in adusted angles (for a unit-area-ellipse) of,

$$\begin{aligned}
\tilde{\theta}_1 \quad &= \quad \tan^{-1}\left(\frac{a}{b} \cdot \frac{y_1}{x_1}\right) = \tan^{-1}\left(\pi a^2 \tan(\theta_1)\right) \\
\tilde{\theta}_2 \quad &= \quad \tan^{-1}\left(\frac{a}{b} \cdot \frac{y_2}{x_2}\right) = \tan^{-1}\left(\pi a^2 \tan(\theta_2)\right).
\end{aligned}$$

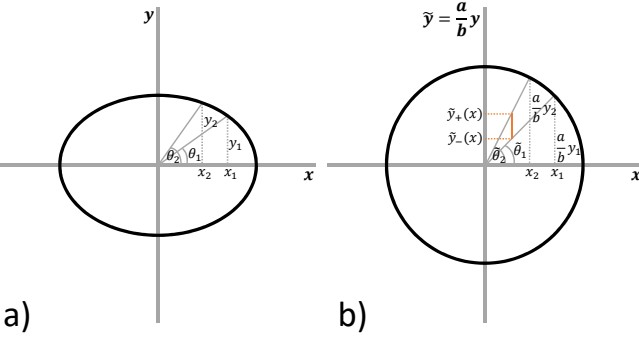

**Figure 1.** Ellipse (a) and scaled-ellipse (b) geometry. Note, in the region $x_2 < x < x_1$, $\tilde{y}_+(x)$ is defined as being located on the circle itself.

The above equations hold for pairs of segment angles falling within the first quadrant of the ellipse. The area of the circle segment in scaled axes is simply,

$$\pi a^2 \left( \frac{\tilde{\theta}_2 - \tilde{\theta}_1}{2\pi} \right) = \frac{1}{2} a^2 \left( \tilde{\theta}_2 - \tilde{\theta}_1 \right). \tag{1}$$

Observing that segment areas may, in general, be written in the form (see Figure 1),

$$\int_0^{x_1} \left( \tilde{y}_+(x) - \tilde{y}_-(x) \right) dx,$$

it readily follows that scaling the y-axis directly scales the calculated area. As such, the area bounded by $\theta_1$ and $\theta_2$ within the unit-area-ellipse (in the original coordinate system) is given by,

$$
\begin{aligned}
A_{\theta_1,\theta_2} &= \frac{b}{a} \cdot \frac{1}{2} a^2 \left( \tilde{\theta}_2 - \tilde{\theta}_1 \right) \\
&= \frac{1}{2\pi} \Big( \tan^{-1} \left( \pi a^2 \tan(\theta_2) \right) \\
&\quad - \tan^{-1} \left( \pi a^2 \tan(\theta_1) \right) \Big).
\end{aligned}
\tag{2}
$$

### 2.3 An elliptical wind direction rose

As outlined above, an ellipse of unit area may be naturally interpreted as a wind direction rose, with the probability associated with any direction segment equal to that segment's area - all of which is determined by the single parameter, $a$. Consider the case where wind directions bins are centered on values $\theta_{c,i}$ (including 0 and $\pi/2$ rads) and of width $2\delta\theta$, where $i = 1, \ldots, 2\pi/2\delta\theta$.

The *elliptical wind direction rose* with elliptical parameter $a$ is defined as,

$$P_{\text{el}}(\theta_{\text{wind}} = \theta_{c,i}, a) = \begin{cases} A_{\theta_{c,i+}, \theta_{c,i-}} & \text{if } \theta_{c,i} > 0 \text{ and } \theta_{c,i} < \pi/2 \\ 2A_{\delta\theta, 0} & \text{if } \theta_{c,i} = 0 \\ 2A_{\frac{\pi}{2}, \frac{\pi}{2} - \delta\theta} & \text{if } \theta_{c,i} = \pi/2 \\ \text{Obtained via symmetries} \\ \text{if } \pi/2 < \theta_{c,i} < 2\pi, \end{cases} \tag{3}$$

where $\theta_{c,i+} = \theta_{c,i} + \delta\theta$ and $\theta_{c,i-} = \theta_{c,i} - \delta\theta$. Note also that $\tan^{-1}\left(\pi a^2 \tan(\pi/2)\right) = \pi/2$. As indicated above, directional probabilities outside the first quadrant may be obtained via symmetries, by reflecting those values across horizontal and/or vertical axes. In computational terms, this amounts to copying, reordering and concatenating the relevant numerical arrays. As a result of these same symmetries, it follows that bins which are opposite (in the context of reflections about the vertical axis) have identical probabilities, i.e.,

$$P_{\text{el}}(\theta_{\text{wind}} = \theta_{c,i}, a) = \begin{cases} P_{\text{el}}(\theta_{\text{wind}} = \pi - \theta_{c,i}, a) \\ \text{if } \theta_{c,i} > 0 \text{ and } \theta_{c,i} < \pi/2, \\ P_{\text{el}}(\theta_{\text{wind}} = 3\pi - \theta_{c,i}, a) \\ \text{if } \theta_{c,i} > \frac{3\pi}{2} \text{ and } \theta_{c,i} < 2\pi. \end{cases}$$

The latter case was written in the above form to maintain all angles between $0$ and $2\pi$. Symmetrical "circular" and "bi-directional" wind roses may therefore be readily obtained by setting only a single parameter, $a$. Via rotation of the resulting elliptical wind rose, one may then set the principal directions in bi-directional cases.

## 2.4 Establishing a prevailing wind direction - the folding parameter

Many sites are not equally bi-directional, and instead have a single prevailing wind direction which dominates. To account for such cases, one may take an elliptical (non-rotated) wind rose and reallocate (i.e,"fold") a certain proportion of probability mass from left-hand-plane segments onto their right-half-plane counterparts. This results in a wind rose with a single prevailing wind direction. Formally, the *generalised elliptical wind direction rose* with elliptical parameter $a$ and folding parameter $f$

80  $(0 \leq f \leq 1)$ is defined as,

$$
P_{\mathrm{g}}(\theta_{\mathrm{wind}} = \theta_{c,i}, a, f) = \begin{cases} (1-f)P_{\mathrm{el}}(\theta_{\mathrm{wind}} = \theta_{c,i}, a) \\ \text{if } \pi/2 < \theta_{c,i} < 3\pi/2, \\[6pt] (1+f)P_{\mathrm{el}}(\theta_{\mathrm{wind}} = \theta_{c,i}, a) \\ \text{if } 0 \leq \theta_{c,i} < \pi/2 \text{ or } 3\pi/2 < \theta_{c,i} \leq 2\pi, \\[6pt] P_{\mathrm{el}}(\theta_{\mathrm{wind}} = \theta_{c,i}, a), \\ \text{if } \theta_{c,i} = \pi/2 \text{ or } 3\pi/2. \end{cases} \tag{4}
$$

The symmetries in the elliptical distribution of directional probabilities guarantee this generalised parametric form is a true probability distributions, i.e. the sum of all directional probabilities remains 1.

As previously, this generalised wind direction rose may be rotated in order to specify a prevailing wind direction, $\theta_{\mathrm{prev}}$.
85  Formally,

$$
P_{\mathrm{g}}^{\dagger}(\theta_{\mathrm{wind}} = \theta_{c,i}, a, f, \theta_{\mathrm{prev}}) = P_{\mathrm{g}}(\theta_{\mathrm{wind}} = \theta_{c,i} - \theta_{\mathrm{prev}}, a, f). \tag{5}
$$

### 2.5   Fitting a generalised elliptical wind direction rose to measured data

Assume we have a vector of wind direction-segment probabilities, $\widehat{\mathbf{P}}$, obtained from measured data and specified for a given set of wind direction bins, centered on values $\boldsymbol{\theta}_c$ (each of width $2\delta\theta$). Further, we assume the circular mean (Dallas et al.,
90  2023) of the measured wind direction data is known. The prevailing wind direction, $\theta_{\mathrm{prev}}$, of the parametric model is set equal to the bin centre-value, denoted $\bar{\theta}_c$, of the bin into which the directional mean falls. Alternatively, $\theta_{\mathrm{prev}}$ may be set equal to the centre-angle of the direction bin with largest probability mass (the mode direction). The best approach in practice was found to be that of fitting both cases, then keeping the one which results in the smallest error[2]. Fitting a generalised elliptical wind direction rose to $\widehat{\mathbf{P}}$ may then be formulated in the context of sum-of-square-errors,

95  $$
\mathrm{SSE}(a, f) = \sum_{i=1}^{2\pi/2\delta\theta} \left( P_{\mathrm{g}}^{\dagger}(\theta_{\mathrm{wind}} = \theta_{c,i}, a, f, \theta_{\mathrm{prev}}) - \widehat{P}_i \right)^2,
$$

specifically, the minimisation thereof with respect to parameters $a$ and $f$,

$$
\min_{a,f} \ \mathrm{SSE}(a, f).
$$

Since $a > 0$ and $0 \leq f \leq 1$, a constrained optimisation would be required for the problem in its current form. However, observing that only $a^2$ appears within the fitting formulation (Equation 2), the constraint on $a$ may simply be removed for the purposes

---

[2]The proposed approach therefore applies the following heuristic: *the $\theta_{prev}$ value resulting in the "best fitting" model is likely either the (circular) mean direction of the dataset, or the highest probability direction; and if neither of these results in a good fit, then a good fit is unlikely to be obtained for any choice of $\theta_{prev}$.*

of optimisation, and then reinstated by taking the absolute value of the result. Unconstrained optimisation which respects the restriction on $f$ may be achieved by setting,

$$f = \frac{1}{1 + e^{-\phi_f}}$$

and performing unconstrained optimisation over $\phi_f$. Partial derivatives may then be obtained, allowing for gradient-descent based optimisation. In the following formulas, $P_{\mathrm{g},i}^{\dagger}$ will denote the probability attributed by the model, $P_{\mathrm{g}}^{\dagger}$, to the $i^{\mathrm{th}}$ direction bin:

$$\frac{\partial \mathrm{SSE}}{\partial a}(a,f) = \sum_{i=1}^{2\pi/2\delta\theta} 2\left(P_{\mathrm{g},i}^{\dagger} - \widehat{P}_i\right) \frac{\partial P_{\mathrm{g},i}^{\dagger}}{\partial a}$$

$$\frac{\partial \mathrm{SSE}}{\partial \phi_f}(a,f) = \sum_{i=1}^{2\pi/2\delta\theta} 2\left(P_{\mathrm{g},i}^{\dagger} - \widehat{P}_i\right) \frac{\partial P_{\mathrm{g},i}^{\dagger}}{\partial \phi_f}$$

$$\frac{\partial A_{\theta_1,\theta_2}}{\partial a} = \frac{a \tan(\theta_2)}{\pi^2 \tan^2(\theta_2) a^4 + 1} - \frac{a \tan(\theta_1)}{\pi^2 \tan^2(\theta_1) a^4 + 1}$$

$$\frac{\partial P_{\mathrm{g},i}^{\dagger}}{\partial \phi_f} = \begin{cases} -f^2 e^{-\phi_f} P_{\mathrm{el}}(\theta_{\mathrm{wind}} = \tilde{\theta}_{c,i}, a) \\ \quad \text{if } \pi/2 < \tilde{\theta}_{c,i} < 3\pi/2, \\[1em] f^2 e^{-\phi_f} P_{\mathrm{el}}(\theta_{\mathrm{wind}} = \tilde{\theta}_{c,i}, a) \\ \quad \text{if } 0 \le \tilde{\theta}_{c,i} < \pi/2 \text{ or } 3\pi/2 < \tilde{\theta}_{c,i} \le 2\pi, \\[1em] 0, \\ \quad \text{if } \tilde{\theta}_{c,i} = \pi/2 \text{ or } 3\pi/2. \end{cases}$$

where $\tilde{\theta}_{c,i} = \theta_{c,i} - \theta_{\mathrm{prev}}$. The partial derivatives $\partial P_{\mathrm{g},i}^{\dagger}/\partial a$ are readily obtained using $\partial A_{\theta_1,\theta_2}/\partial a$ (see Equations 3-5). The full expression is not included here for the sake of brevity.

### 2.5.1 Assessing goodness-of-fit

A coefficient of determination, typically denoted $R^2$, may be calculated for the resulting fit to measured data:

$$R^2 = 1 - \frac{\sum_{i=1}^{2\pi/2\delta\theta} \left(P_{\mathrm{g}}^{\dagger}(\theta_{\mathrm{wind}} = \theta_{c,i}, a, f, \theta_{\mathrm{prev}}) - \widehat{P}_i\right)^2}{\sum_{i=1}^{2\pi/2\delta\theta} \left(\widehat{P}_i - \overline{\widehat{\mathbf{P}}}\right)^2},$$

with $\overline{\widehat{\mathbf{P}}}$ the mean of all measured directional probabilities. This value falls between 0 and 1 and describes the proportion of total variance, present in the measured data, captured by the fitted model. The root-mean-squared-error may also be calculated when assessing goodness of fit,

$$\mathrm{RMSE} = \sqrt{\frac{2\delta\theta}{2\pi} \sum_{i=1}^{2\pi/2\delta\theta} \left(P_{\mathrm{g}}^{\dagger}(\theta_{\mathrm{wind}} = \theta_{c,i}, a, f, \theta_{\mathrm{prev}}) - \widehat{P}_i\right)^2}.$$

Considering the above, one may observe that these measures provide both a normalised $\left(R^2\right)$ and an absolute (RMSE) measure for goodness of fit. It is highlighted that, as a consequence, the RMSE-scale is dependent on the chosen number of direction bins.

## 3 Results

Example generalised elliptical wind direction roses are shown in Figure 2, along with their associated parameter values. This demonstrates the flexibility of the proposed parametric model. Each direction rose may be rotated to obtain any required prevailing wind direction (or directions, when $f = 0$). Results from fitting the parametric model to real wind direction rose data are shown in Figure 3. These example cases include direction-bin sizes of $5°$, $15°$ and $30°$, which together span the standard range seen in practice (as a point of reference, IEC 61400-12-3:2002 specifies a direction-bin size of $10°$ during measurement-based site calibration of power performance). A good fit is achieved in most cases, indicating the proposed model is representative of real wind direction roses. Goodness-of-fit values highlight the fact that the RMSE-scale is dependent on the number of wind direction bins. Since $R^2$ values are normalised, they can be seen to provide a strong indication of model fit-quality that is independent of the number of bins. Limitations of $R^2$ should be kept in mind when utilising it to assess goodness-of-fit[3] (Hahn, 1973; Barrett, 1974). However, in the current case the aim is not that of producing a predictive model, hence, the main limitations of $R^2$ are unlikely to be of significance here. It is interesting to note that the various fitted parametric models include both mean and mode $\theta_{\text{prev}}$ cases, as well as cases where the two coincide. Finally, results highlight that a good fit to measured data will not be achieved in all cases (Figure 3e), but, such instances will be flagged by low $R^2$ scores.

## 4 Discussion

The results presented in the previous section indicate the presented parametric model will likely prove useful to wind energy research and development efforts. To support such utility, the current section seeks to clarify the intended use cases for the model, and to outline opportunities for its extension.

Beginning with the former, it is highlighted that the generalised elliptical wind direction rose, $P_g^\dagger$, is proposed as a direction-distribution counterpart to the Weibull wind speed distribution; *i.e.* it captures the general shape of typical wind direction distributions within a simple model with a small parameter-set. As outlined in Section 1, such a model provides an opportunity for standardisation, the identification of typical parameter ranges, and sensitivity analyses concerning direction rose shape impacts. As such, the proposed model is expected to provide utility in generalised studies which consider a range of possible site conditions (for example, when evaluating a proposed innovation's broad efficacy or when seeking to elucidate fundamental relationships or mechanisms within wind farms). If, on the other hand, an analysis is being undertaken for a known site with empirical wind rose, there is no need for a parametric model. In such cases the site's empirical wind rose should be used directly. A possible exception to this is when undertaking complex turbine-layout optimisations at specified sites. The

---

[3]$R^2$ alone does not determine how well a fitted model may be used for prediction. In effect, it therefore only *partially* measures the usefulness of the fitted model (Hahn, 1973; Barrett, 1974).

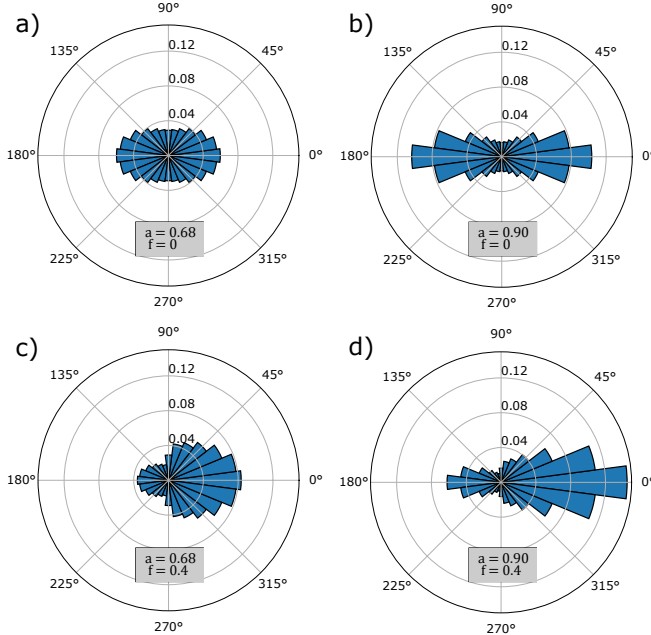

**Figure 2.** Example generalised elliptical wind direction roses, showcasing the various forms the parametric model can generate. Parameter values are given in each case.

proposed direction-rose model could provide a smoothed version of a site's empirical wind rose to facilitate a fast first-pass optimisation, the result of which may then be used to initialise a second-round optimisation using the site's full empirical wind rose. Reductions in overall optimisation times for such analyses might therefore be supported by the proposed model.

Secondly, it is pertinent to consider how the current direction rose model could be extended. One might consider alternative approaches to "folding" the ellipse. For example the current discontinuity, when transitioning between $1-f$ and $1+f$ regions (Equation 4), could be smoothed by linearly scaling from $1-f$ to $1+f$ about the ellipse circumference (or similar). Another possibility is the addition of a second folding parameter, which acts to fold the distribution across the other (semi-major) axis of the ellipse. Improved fitting might also be achieved via inclusion of the prevailing wind direction ($\theta_{\text{prev}}$) within the optimisation procedure; however, this would require the fitting procedure to be extended to ensure robustness against local minima. The heuristic approach utilised in the current work, to determine prevailing wind direction, was incorporated to circumvent this very issue. Finally, mixture models which linearly combine two or more generalised elliptical wind direction roses (scaled by proportional weightings which sum to one) allow for more complex multi-modal wind direction rose realisations, see Figure 4. Many of these outlined model extensions require an expansion of the model parameter-set, which in turn increases the complexity and difficulty of model fitting. Additionally, there is an important balance to be struck between generality and specificity in this context. While increasingly complex versions of the proposed model would certainly fit ever closer to empirical wind rose datasets, the utility associated with having a small parameter-set (representative cases, parameter ranges,

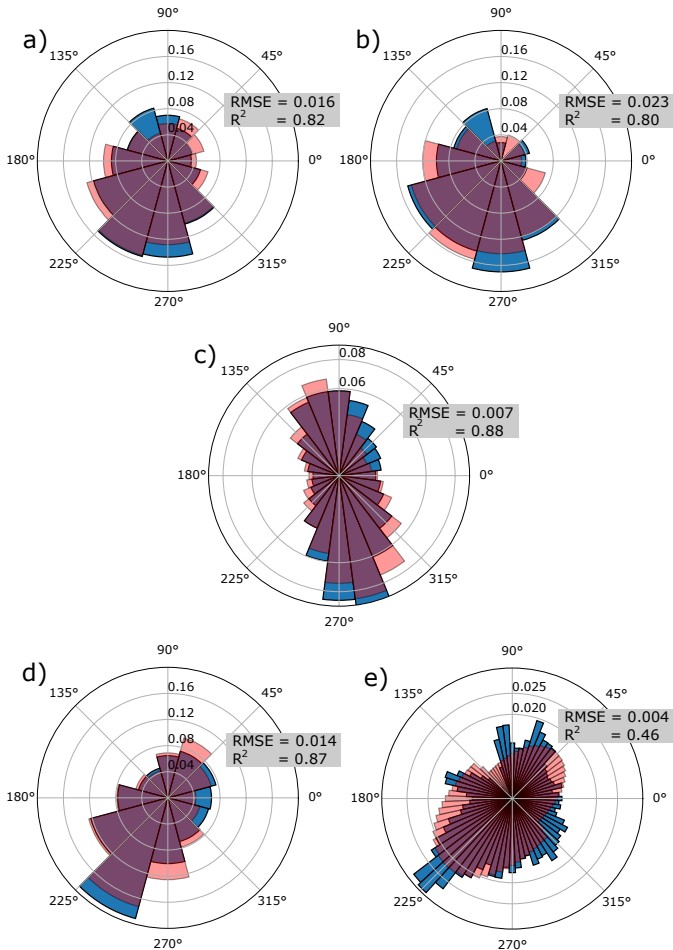

**Figure 3.** Wind direction roses as measured at real offshore wind farms (blue), and the corresponding best-fit generalised elliptical wind direction model (red) in each case. Goodness-of-fit measures are also provided. Wind farms include a) Horns Rev 1 (Pedersen et al., 2023) b) Lillgrund (Pedersen et al., 2023) d) Borssele (Kainz et al., 2024) and e) Princess Amalia (Python Wind Rose). The wind rose in c) is from (King et al., 2021).

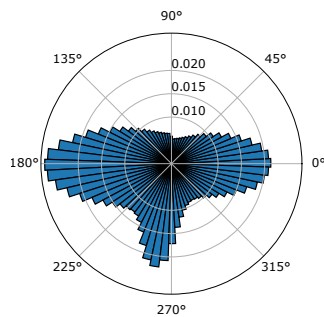

**Figure 4.** Example mixture model which combines two generalised elliptical wind direction roses to generate a tri-modal distribution. The constituent parametric models have parameter values of $(a_1, f_1, \theta_{\text{prev1}}) = (0.8, 0.1, 180°)$ and $(a_2, f_2, \theta_{\text{prev2}}) = (1.1, 1, 260.5°)$ and summation coefficients of 0.85 and 0.15, respectively.

standardisation) could also be undermined or completely lost. Careful consideration should therefore be afforded to these competing factors.

## 5 Conclusions

A parametric model for the wind direction rose has been presented, based on ellipse geometry and then extended to allow a prevailing wind direction to be established. Relevant equations were developed to allow the parametric model to be fitted to measured data, via gradient descent minimisation of sum-of-square-errors. Testing on real offshore wind farm data indicated the parametric model is indeed representative. $R^2$ and RMSE goodness-of-fit measures were utilised, with the former providing a strong indication of model suitability which is independent of the number of direction bins. It was highlighted that the proposed parametric model will not always provide an accurate fit, but that the $R^2$ value should flag when this is the case. Overall, presented results indicated good performance for the proposed direction rose model. Use cases and possible model extensions were then discussed, including the need to balance generality with specificity when undertaking any such extension. It should be appreciated that the proposed model describes wind direction distributions only, and is independent of wind speed. Depending on context, a single Weibull wind speed distribution may be applied across all bins, separate Weibull distributions may be defined for the "prevailing" and "non-prevailing" half circles, or a different Weibull distribution may be defined for each bin. It is hoped the presented parametric model will prove valuable to the wind industry, by providing an opportunity for standardisation and enabling systematic analyses of wind direction distribution impacts and sensitivities for proposed wind farm design and control innovations.

*Code and data availability.* A demonstrator Python implementation of the proposed model, along with all data used in this brief communication, can be accessed via the following link: https://doi.org/10.15129/cb41576e-892e-4415-a849-9af38bc6ad1d .

*Author contributions.* EH undertook all aspects of research and paper preparation.

185 *Competing interests.* The author declares there are no competing interests.

*Acknowledgements.* The author would like to thank Julian Feuchtwang for originally proposing ellipse geometry as a route to wind direction rose modelling, and Julian Quick for stimulating discussions and assistance in tracking down wind rose data. The author is also grateful to the two anonymous reviewers for their insightful comments and suggestions. Edward Hart is funded by an EPSRC Innovation Launchpad Network+ Researcher in Residence Fellowship (EP/W037009/1), a collaboration with the Offshore Renewable Energy Catapult.

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
