# Peer review of "Brief communication: An elliptical parameterisation of the wind direction rose"

_Wind Energy Science, 2024_

## Author Response (AR1)

**Brief communication: An elliptical parameterisation of the wind direction rose (wes-2024-187)**

**Response to Reviewer 1**

Dear Reviewer,

Thank you for taking the time to consider this submission to WES. I am grateful for your insights, and your helpful comments and suggestions. I will revise the manuscript in-line with your suggestions, and I believe it will be much improved as a result.

Please find my detailed responses below, where I also include your comments in blue.

Additional comments have now been added in **RED**, these indicate the post discussion revisions which have now been made to the manuscript.

This paper presents a parametric wind direction rose model based on an ellipse and demonstrates how the model, which includes 3 parameters, can be fit to measured wind direction data for a variety of sites. There is little published on parametric wind direction rose models, so this is a novel contribution to the literature. When combined with wind speed distributions, the model could potentially serve as a standard wind rose definition for computing wake losses, lifetime loads from wakes, and wind farm control benefits for a wind farm, similar to how the Weibull distribution is used to model wind speed probabilities for annual energy production and fatigue load calculations for individual turbines. The parameters of the elliptical wind direction rose model could be used to standardize the characterization of wind direction distributions in the wind industry. They could also be varied to explore the sensitivity of wind plant performance, loads, and wind farm control strategies to different wind direction distributions.

I am glad you agree this is a novel contribution to the field, with a number of potential applications.

The main comment I have is about how the elliptical parameterized wind rose can be made more useful for sites with more complex wind direction distributions. The presented parameterization works well for sites with unimodal wind roses or bimodal wind roses where the prevailing wind directions are in opposite directions, as shown in Fig. 3. However, similar to what is shown in Fig. 3e, there are many sites where the most common wind directions are not 180 degrees apart. To provide another example, in Fig. 3 of Bensason et al. 2021 (https://pubs.aip.org/aip/jrse/article/13/3/033303/285076/Evaluation-of-the-potential-for-wake-steering-for), the most common wind direction sectors are from the northwest and south. It would greatly strengthen the paper to discuss possible extensions of the elliptical wind rose model that could more accurately describe these types of wind direction distributions. For example, could you consider a linear combination of elliptical wind roses with different prevailing wind directions such that the sum of the distributions integrates to 1? This could be a nice solution if the different prevailing wind directions could be included as optimization parameters, rather than being identified manually. Of course the idea behind a parameterized wind rose model is to keep it relatively simple, but considering only wind roses with prevailing wind directions 180 degrees apart might be too much of a simplification for many sites. Among other issues, this could be important when estimating wake losses at a site with long distances between rows of turbines but close spacing in the perpendicular direction. Underrepresenting the likelihood of common wind directions aligned with the close turbine spacing in order to fit the wind rose model to the prevailing direction might cause wake losses to be significantly underestimated.

You raise an excellent point here, and I agree that this brief communication would be improved by discussing/outlining possible extensions to the more complex cases you describe. While developing the parametric wind direction rose model (I'll refer to this as PWDM) I did indeed consider the opportunity to extend to a PWDM mixture-module by taking a weighted linear combination, and constraining the weights to sum to 1. A benefit of this approach would be that more complex sites could be represented, a drawback would be the associated uplift in the numbers of parameters to be fitted via optimisation (both of which you identified in your comment). Having said that, most mixture-model cases would likely only require two PWDMs to be included, and one might simplify further by assuming the prevailing wind directions for each might be set manually (which still allows for standardisation and sensitivity analyses, if not complete automation). Irrespective of the particular implementation, I agree these possible extensions should be discussed in the manuscript. I will therefore add a new section on "Mixture-model extensions for more complex sites". I have now added a Discussion section (Section 4) which describes these possible model extensions and others. An example mixture model is included there (Figure 4).

Comments:

1. Pg. 3, ln. 60: Is this equation only supposed to be valid when theta^tilde_1 and theta^tilde_2 are less than or equal to pi/2? If so, please clarify. Also, looking at Fig. 1b, how is y^tilde_+(x) defined for x_2 < x <= x_1? The segment area is no longer bounded by two lines like it is for x < x_2.

Yes you're right that the equation is valid (and later applied) only where both angles are less than or equal to pi/2, and you're right that this should be clarified here. The following clarification has been added below the equations for theta_tilde_1 and theta_tilde_2: *"The above equations hold for pairs of segment angles falling within the first quadrant of the ellipse."*

In the queried region, y+ is determined by the circle itself. This should be clarified in the manuscript also, thanks! This has been clarified in the caption of Figure 1.

2. Pg. 4, ln. 82: "not be equally" -> "not equally"

Will fix, thanks! Sorted!

3. Section 2.4: Scaling the wind direction probability by 1-f for pi/2 < theta_c,i < 3pi/2 and 1+f for 0 <= theta_c,i < pi/2 or 3pi/2 < theta_c,i <= 2pi causes a sharp discontinuity at theta_c,i = pi/2 and 3pi/2, which doesn't seem very realistic. Would a smooth (e.g., linear) transition from 1-f to 1+f be more appropriate? One example would be scaling P_el by (1 + f - (2*f/pi)*theta_c,i) for 0 <= theta_c,i < pi. This way you would still get 1 - f for theta_c,i = pi, 1 for theta_c,i = pi/2 and 1 + f for theta_c,i = 0.

This is an interesting point you raise. My thoughts are as follows:

1) The discontinuity in scaling occurs across the smallest probability bins of the elliptical wind rose (with symmetrically larger bins either side). As a result the outcome of this tends to be a scaled wind rose in which the bin probabilities increase fairly smoothly across these points. In many cases there is therefore no clear discontinuity present for the final parametric wind rose. But you are right that a more marked discontinuity will appear at small bin sizes, even given the above factors.

2) While a discontinuity is introduced in the scaling factor as you describe, and in many scenarios discontinuities can be a problematic, those problematic cases tend to be where the rate-of-change or smoothness of a function has a bearing on the result. For a wind rose we're simply representing the probabilistic "weight" associated with each direction bin, independently of the others (barring the restriction that the total probability must be 1). I'd argue that a discontinuity therefore isn't *intrinsically* a problem here, and that the key question is more along the lines of: "do we produce realistic looking wind roses and/or obtain a good fit to real data?" The current model seems to allow us to answer yes to both of these, and so I don't believe the highlighted discontinuity is a major issue or impediment to the usefulness of this model at this stage.

3) Having said that, alternative "folding" parameterisations (such as the linear one you suggest) would open up new shape variations and could provide superior fits in some cases. Therefore, while I don't believe there is an immediate need to alter the original folding approach, I believe there is value in the possibility for alternative approaches being described in manuscript, supporting more flexible implementations within future work. I will therefore include such a discussion when revising the manuscript, and I thank you for this valuable suggestion!

The possibility of using a smoother folding approach is now included in the newly added Discussion section (Section 4, line 153).

4. Section 2.5: Could you also optimize the prevailing wind direction theta_prev when fitting a wind rose to empirical data?

You could indeed include the prevailing wind direction as an optimisation parameter, however I believe local minima would become a problem when trying to ensure robust parameter identification. Another option would be to exhaustively test each possible direction, but this feels inelegant as a solution. Instead I worked with the heuristic that the "best" prevailing wind direction for the model would likely be either the highest probability direction, or the (circular) mean direction, and that if neither of these resulted in a good fit (indicated by high $R^2$) then a good fit was unlikely to be obtained using any direction. This heuristic has worked well across all cases tested thus far. There is certainly room for this aspect of model fitting to be explored in more detail, and so I will include the above discussion when updating the manuscript. A clearer description of the utilised prevailing wind direction heuristic has been added to Section 2.5 (line 92 + footnote). Additionally, the possibility of optimising the prevailing wind direction alongside other parameters has been added to the new Discussion section (Section 4, line 156).

5. Pg. 6, equation after line 110: in the first two lines, it would clarify the equation if "i" were added as a subscript for P^dagger_g because this represents the probability of the specific bin "i".

Agreed, I'll add that! This has been added as suggested.

6. Pg. 6, ln. 115: "The partial derivative del P^dagger_g / del a is readily obtained using del A_theta_1,theta_2 / del a…": To help the reader, it would be good to refer to the specific equations earlier in the text that show how these two partial derivatives are linked. This might require more equations to be numbered.

Good point, I'll do that! Done as suggested, including additional equation numbers.

7. Section 3: It would be helpful to discuss the choice of bin sizes shown. What are typical wind direction bin widths for wind roses in the wind industry, for example for energy yield assessments or controls analysis?

Agreed. In my experience it's from 5 to 30 degrees, but I'll double check the certification requirements so I can add that helpful additional context. Clarification of this point has been added, including information on the direction-bin size specified in IEC 61400-12-3 for power performance calibration. See Section 3, line 126.

8. Pg. 7, ln. 130: "the RMSE-scale is dependent on the number of wind direction bins." Couldn't the RMSE be normalized to account for the number of wind direction bins so it can be used to compare the goodness-of-fit for roses with different bin sizes?

Yes, this is essentially what $R^2$ is doing for us. I'll point that out when revising the manuscript. The reason why it's nice to have both is that it ensures one has a normalised ($R^2$) and absolute (RMSE) measure for goodness of fit. This has been done as described (line 119).

9. Pg. 7, ln. 131: "Limitations of $R^2$ should be kept in mind" Please briefly discuss these limitations here.

I'll add some details to the manuscript as you suggest. This will also allow me to indicate more explicitly why those limitations aren't a major issue in this application. A footnote has been added here which elaborates on these details (Section 3, footnote 3).

Thanks again,

Edward Hart

Senior Lecturer // Chancellor's Fellow
Wind Energy and Control Centre
Dept. of Electronic and Electrical Engineering
The University of Strathclyde
Glasgow, UK

**Brief communication: An elliptical parameterisation of the wind direction rose (wes-2024-187)**

**Response to Reviewer 2**

Dear Reviewer,

Thank you for taking the time to consider this submission to WES. I am grateful for your insights, comments and suggestions, and I believe the manuscript will be much improved as a result. Please find my detailed responses below, where I also include your comments in **blue**.

Additional comments have now been added in **RED**, these indicate the post discussion revisions which have now been made to the manuscript.

The manuscript proposes a parametric model for the probability of observed wind directions, and it might be used for any circular probability distribution. The aim is to provide a smooth wind direction rose, suitable for optimization of wind farm layouts or advanced wind farm control.

I would disagree with this characterisation of my aim in developing this parametric wind direction rose model. I would instead state the aim as being *the provision of a simple parametric direction rose model which supports standardisation and the identification of generally representative cases, and enables systematic sensitivity analyses of wind rose "shape" impacts on wind farm innovations*. I have, therefore, set out to develop a model which effectively captures the general shape of typical wind direction roses, in much the same way a 2-parameter Weibull distribution captures the general shape of annual mean wind speed distributions. Crucially, neither the Weibull distribution nor the presented parametric wind rose model have much utility if one is working to optimise layout or control at a single real site. In such cases one should simply use the empirical distributions of wind speed and wind direction for the known site. Rather, I believe the presented model has important utility in cases where one is focussed on developing a capability or technique which might then be applied to a variety of sites, and/or if one is seeking to investigate fundamental relationships between site characteristics (including wind direction rose shape) and reliability or yield impacts. An example of the former would be in the development of optimisation tools for layout and control co-design (in which wake effects, and so the wind direction rose, play an important role). In order to demonstrate efficacy for such tools, and motivate their ongoing development or real world application, the potential benefits will generally be quantified for a theoretical wind farm, typically using only a single arbitrarily selected wind rose. If, for example, 1% more power is shown to be generated as a result, that provides some quantification of the potential benefits. However, it is also unclear how much that value might change between sites with more uniform wind roses, versus strongly bi- or uni-directional wind roses. Utilising the presented parametric model, a detailed sensitivity analysis of the wind rose shape impacts on yield may now be undertaken to provide both an improved characterisation of potential benefits (e.g. yield might in-fact vary between 0.8% and 3.4% based on the shape of wind direction rose) and an enhanced conceptual understanding of the problem (e.g. co-design benefits may not be worth pursuing for more uniform wind roses with elliptical parameter below a given value). These same benefits would also hold in analyses of wake impacts on turbine subcomponent reliability, where the goal is not that of characterising reliability impacts for a single real site, but instead to provide a general and fundamental improvement in our scientific understanding of these effects across sites of different types. Beyond this, the parametric direction rose model provides an opportunity to standardise our characterisation of wind rose "shape" and identify normal parameter ranges across which sensitivity analyses should be considered, again in

much the same way that the Weibull distribution is used to characterise wind speed distributions. To help clarify the intended use cases for the model, I have added a Discussion section in which this is now outlined (Section 4, paragraph 2). The possibility for the model to provide a smoothed version of an empirical wind rose is also included there (line 148).

The basic model takes the shape of an ellipse, and, to allow more flexibility, it folds part of the probability mass in half of the ellipse upon the opposite half. An expression for the area of an ellipse sector is presented and used to fit the parametric model to observed wind sector frequencies. The model-fitting principle is the minimization of the sum of squared errors. For this purpose, the author presents equations for derivatives of the objective function with respect to model parameters. The model does not yet include a directional variation of the wind speed distribution.

I would counter the claim that "The model does not yet include a directional variation of the wind speed distribution". Model fitting does include a determination of the prevailing wind direction, based on the data being fitted to. You are indeed correct that this aspect of modelling fitting is not part of the optimisation scheme. The prevailing wind direction is instead identified by fitting parametric models to the data assuming a prevailing wind direction of a) the circular mean direction and b) the highest probability wind direction bin, before keeping the one which provides the best fit. This heuristic is based on the logic that if neither of these prevailing wind directions produce a good fit to the measured data, then the model is unlikely to result is a good-fit for others. That logic has borne out through testing. While the prevailing wind direction could become an additional optimisation parameter, it would likely introduce problems related to local minima into the optimisation. As the current formulation performs well, and the heuristic has stood up well, I see not immediate need to embed prevailing wind direction within the optimisation itself. Having said that, the above points should probably be more clearly discussed within the paper, and so I will elaborate on these points when revising the manuscript. A clearer description of the utilised prevailing wind direction heuristic has been added to Section 2.5 (line 92 + footnote). Additionally, the possibility of optimising the prevailing wind direction alongside other parameters is considered in the new Discussion section (Section 4, line 156).

Gradient-based layout optimization algorithms will accept larger wake effects in sectors with low frequency of occurrence and thereby smaller contributions to annual energy production. If the input wind rose is too detailed, the algorithm's convergence may be slow, and the solution will be sensitive to random variations. Thus, models with smoother directional variation are needed for some purposes. On the other hand, the wind-rose simplification should not significantly alter the predicted energy production with or without wake correction. At most sites, the wind speed distribution depends on direction, so we risk that the energy production estimate changes after modifications of the wind rose.

These are all valid and important points. As detailed in my first (long) response comment, I am not proposing that this model be applied in cases where design is being undertaken for a single known site. In such cases the empirical distribution should simply be used directly. But you also highlight here a potential application for the parametric model that I'd not previously considered, that of possibly providing a smoother representation of a site's wind rose in order to facilitate a faster first-pass optimisation. That result could then provide a first guess for initialising a second optimisation in which the actual (non-smooth) wind rose is reintroduced. There would of course be caveats to this, such as those you outline. I will seek to include a discussion of both the potential opportunity and the caveats when revising the manuscript. I will also highlight the ever important point that, much like for a Weibull distribution, there will be instances in which more detail is required and so a simplified

representation is not suitable. This additional potential application is now included in the new Discussion section (Section 4, line 148).

An ellipse is symmetrical over both major and minor axes, so we might fold over either or both of them. Just remember that the rotation angle should be included as an optimization parameter if we choose to fold over both axes. Unfortunately, the fold-over procedure introduces discontinuities in the dictations along the minor axis, which might reintroduce the disadvantages of the raw wind rose.

I agree you could fold over both axes if looking to extend the model to be more flexible. Having said that, the necessity of increasing the number of model and optimisation parameters by 2 (additional fold + prevailing wind direction) makes we feel the benefits of a simple parametric model might start to be lost, in addition to furthering complicating the optimisation with local minima. But, it is certainly a valid point and I will make sure to include this observation in the revised manuscript. On the point concerning discontinuities, you are right this might slow layout optimisations if smoothness is a primary goal of the parametric model. However, as outlined earlier in my response, smoothness was not the primary motivating factor behind the proposed model. This additional potential application is now included in the new Discussion section (Section 4, line 154).

The model is fitted by a raw wind rose with discrete sector statistics, but it might be more accurate to fit directly by data. The result seems to be a new sector-based distribution, but working with the underlying continuous distribution in optimization algorithms might be better.

The proposed model was developed with a focus on simplicity and easy applicability, which is why it fits directly to the wind rose, rather than raw wind data. Importantly, I believe this still allows all of the principal *aims* (detailed in my first long response comment) of the model to be fulfilled. In addition, there is the added benefit that, in practice, site wind roses are most commonly available in the form of sector probabilities rather than raw wind data.

The von Mises (vM) distribution is the classic model for circular statistics. Due to its unimodal distribution, it is rarely used in wind engineering, but the generalized von Mises distribution (GvM) supports an arbitrary number of modes. GvM models are not easy to fit to data, but Kim and SenGupta present a promising numerical method based on the maximum likelihood principle, see https://doi.org/10.1080/02664763.2020.1796938. The book "Directional Statistics" by Mardia and Jupp discusses more options, see https://onlinelibrary.wiley.com/doi/book/10.1002/9780470316979.

Thank you for highlighting these resources. While there are other approaches to characterising directional distributions, as you correctly point out, they tend to be tricky to fit and can have large numbers of parameters. Additionally, the fitting itself can require a strong knowledge of statistical theory to grasp. Instead of going down that route, I have developed a geometrically driven model which, with a small number of parameters and straightforward sum-of-squares-error fitting, is shown to successfully capture the general shape of various real wind roses. As such, I believe there is significant benefit to the parametric model is have presented. I have now included these additional references in the introduction (Section 1, line 20), where I highlight that existing circular statistical models are both complex and difficult to fit. Thank you for bringing these additional relevant citations to my attention.

I once used a more straightforward approach, fitting Fourier splines to the observed sector frequencies and directional variations of the mean and cube of the wind speed. A low-pass filter in wave number domain provided flexible directional smoothing, and Weibull

distributions for wind speed from different directions were derived by statistical moments of the wind speed.

This does sound interesting as an approach, but also not very generalizable (i.e. there isn't a small number of parameters which represents any individual wind rose, such that you can say "most wind roses have the following parameter ranges" etc – which I see as a key benefit of my proposed model). I'd suspect you were aiming to smooth the wind direction rose for improved performance in layout optimisation (or similar)? As described above, while that's certainly a valuable capability, it was not my particular aim when developing this model.

I suspect that the new elliptical model offers too little flexibility for accurate wind farm production estimates. However, it might be useful for special purposes like the development of wind farm control strategies or fast approximate layout optimization.

As previously described, accurate production estimates for an individual wind farm would indeed not benefit from the proposed model, and instead the empirical distribution should simply be used instead. As you then observe, the parametric model is instead mostly directly conceived as a valuable tool when developing (and exploring potential benefits of) a given capability or technique which might then be applied to a variety of sites, and/or if one is seeking to investigate more fundamental relationships between site characteristics and performance or subcomponent reliability impacts. Intended use cases for the proposed model are now clarified in the new Discussion section (Section 4, paragraph 2). It is also pointed out there that if only a single site is under consideration, then the empirical wind direction data should simply be used directly.

Finally, concerning flexibility, more complex wind rose representations can be readily obtained by extending the proposed parametric model to a mixture of such models, allowing for multi-modal wind roses. A drawback of this would be increased numbers of parameters during optimisation, but one could go down a middle-road by requiring the prevailing wind direction of each mixture component to be manually specified. Anyway, within the revised manuscript I will discuss these possibilities and provide an example of a two-model mixture wind rose to highlight that further development towards more general cases is very possible. A range of possible model extensions are now described in the new Discussion section (Section 4, paragraph 3), including their benefits and potential drawbacks. An example mixture model is also included (Figure 4).

SPECIFIC COMMENTS:

P1, line 15: I was puzzled by the expression "energy uplift obtained for a single candidate wind rose". Try to reformulate for clarity.

Will do! I have now changed "energy uplift" to "increased energy yields" to improve clarity here.

P2, line 36-48: The explanation of the eccentricity is not used in model formulation, so it might be left out.

I included it for completeness, but will reconsider whether to include or not. On reflection, I agree that the discussion of eccentricity is unnecessary and distracting, hence I have now removed those parts of the manuscript.

Section 2.3: The multi-case equations in this paragraph are complex to read. Maybe you could simplify by using the Arg or Atan2 functions.

I agonised over the best way to present these equations when developing the paper and this I found to be best. I think there will be multiple cases whichever way the formulation occurs, because of direction bins straddling quadrants. In addition, the presented formulation matches the code implementation (which will be made available alongside the final published paper), and so for consistency reasons I think the current presentation is likely best. Having said that I will take another look to make sure! On reconsidering these points I have decided to stick with the existing formulations, as they leverage the natural symmetry present on the model and also match the code implementation I am releasing alongside the paper.

Thanks again,

Edward Hart

Senior Lecturer // Chancellor's Fellow
Wind Energy and Control Centre
Dept. of Electronic and Electrical Engineering
The University of Strathclyde
Glasgow, UK

---

## Referee Report (RR1)

The revised article wes-2024-187-ATC2 includes corrections by both reviewers. The new discussion section is an improvement, which contemplates various model extensions and clarifies the scope of the model.

The paper is nearly ready for publication. The only remaining issue is to update the link in the paragraph called 'code and data availability'.

Thanks for the well-structured author-response letter.

---

## Author Response (AR2)

Dear Alfredo and Jakob,

I am happy to confirm that the link to code and data in the paper is now live.

Many thanks,

Edward Hart